# Regulation Effects of Beeswax in the Intermediate Oil Phase on the Stability, Oral Sensation and Flavor Release Properties of Pickering Double Emulsions

**DOI:** 10.3390/foods11071039

**Published:** 2022-04-02

**Authors:** Meimiao Chen, Wenbo Wang, Jie Xiao

**Affiliations:** 1Guangdong Provincial Key Laboratory of Nutraceuticals and Functional Foods, College of Food Science, South China Agricultural University, Guangzhou 510642, China; mm15813387110@163.com (M.C.); abdullah_ch2002@yahoo.com (A.); 2College of Electronic Engineering, South China Agricultural University, Guangzhou 510642, China; wwb@scau.edu.cn

**Keywords:** Pickering double emulsions, gelation, beeswax, stability, oral sensation, flavor release

## Abstract

Double emulsions (W/O/W) with compartmentalized structures have attracted a lot of research interests due to their diverse applications in the food industry. Herein, oil phase of double emulsions was gelled with beeswax (BW), and the effects of BW mass ratios (0–8.0%) on the stability, oral sensation, and flavor release profile of the emulsions were investigated. Rheological tests revealed that the mechanical properties of double emulsions were dependent on the mass ratio of BW. With the increase in BW content, double emulsions showed a higher resistance against deformation, and lower friction coefficient with a smoother mouthfeel. Turbiscan analysis showed that the addition of BW improved the stability of double emulsions during a 14 days’ storage, under freeze–thawed, and osmotic pressure conditions, but it did not improve the heating stability of double emulsions. The addition of BW contributed to lower air-emulsion partition coefficients of flavor (2,3-diacetyl) compared to those without the addition of BW at 20 °C and 37 °C, respectively. Furthermore, the addition of BW and its mass ratio significantly altered the flavor release behavior during the open-bottle storage of double emulsions. The response value of 0% BW dropped sharply on the first day of opening storage, showing a burst release behavior. While a slow and sustained release behavior was observed in double emulsions with 8.0% BW. In conclusion, gelation of the intermediate oil phase of double emulsions significantly enhanced the stability of double emulsions with tunable oral sensation and flavor release by varying the mass ratio of beeswax.

## 1. Introduction

Double emulsion, also known as multiple emulsion, is a type of emulsion system, which has emulsion drops that contained many smaller droplets inside, including W/O/W and O/W/O double emulsions [1,2]. Due to their multi-chamber or compartmentalized internal structures, double emulsions (W/O/W) have emerged as ideal delivery systems for the encapsulation and co-encapsulation of flavors, hydrophilic and hydrophobic bioactives, probiotics, etc. [3,4,5]. Since the internal water phase can replace part of the oil phase, double emulsions exhibit potential in the development of low-fat foods [6]. In addition, the special two-membrane and three-phase compartmentalized structures in double emulsions impart an ability to delay the release of encapsulated functional ingredients and consequently to achieve a sustained or targeted release [7,8].

Although double emulsions had great potential in product applications, however, the production of stable double emulsions is still a major challenge due to their high interfacial area and thermodynamic instability. The interfacial film fusion such as droplets flocculation, coalescence, and solute exchange may occur during their processing or storage, which then lead to the degradation of O/W emulsions [9]. The gelation of different phases (internal-, intermediate-, and external phase) is a commonly employed strategy in improving the stability of double emulsions. The regulation effects were closely related to the type and concentration of gelling agents, and processing conditions such as pH, temperature, and ionic strength that trigger the gelation process. In recent years, research attempts to improve the stability of double emulsions by gelation of the internal or external aqueous phase using proteins or polysaccharides as the gelling agents had attracted tremendous attentions [10,11,12]. Different mechanisms have been reported, including the significant increase in the viscosity and the formation of three-dimensional networks (3D) effectively retarded or slowed down the droplets movement, thus, inhibition of coalescence.

Lipid-based gelators, such as waxes, fatty acids, and monoglycerides exhibit the capacity of self-assembling in the lipid phase or at the interface to form the crystal lipid network in the lipid phase [13]. Sun, Zhang and Xia [14] used glycerol monostearate (GMS) to form the oleogels and improved the stability of double emulsions through reinforcing oil film. Gelation of the oil phase not only solidified the oil globules, but also reinforced the oil film and formed steric barrier, thereby offered flexible regulation over the stability of double emulsions. In addition, it has been reported that gelation of the oil phase had regulation effects on the texture and flavor release properties in emulsions [15]. Chen, Guo, Wang, Yin and Yang [16] found that the structured O/W emulsion based on the crystallization of β-sitosterol delayed volatiles release under real time dynamic conditions due to the steric barrier effects. Therefore, we have reason to believe that the gelation of the oil phase in double emulsions may not only improve the stability, but also present great potential in controlling the oral sensation and flavor release attributes.

Recently, our team prepared the Pickering double emulsion (PDE) using bacterial cellulose as the external stabilizing particles that resulted in significantly enhanced stability [3]. In this work, we aimed to incorporate an oil phase gelator into the intermediate oil phase to create gelled double emulsions with improved stability and tunable sensory and flavor release profiles. Beeswax, widely used in the food industry as food coating and packaging material due to its plasticity, moisture resistance, and antioxidant properties [17,18,19], was selected as the gelator. The effects of different mass ratios of beeswax on the microstructure were investigated using inverted fluorescence microscopy. We also characterized the stability of emulsions during processing and storage, as well as their tribological and rheological properties via turbiscan and rheometer, respectively. Finally, we studied the release behavior of flavor in terms of the partition coefficient and their retention effects within double emulsions with different mass ratios of BW.

## 2. Materials and Methods

### 2.1. Materials

The Damao Chemical Reagent Factory (Tianjin) provided the analytical grade sodium citrate and citric acid. Polyglycerol polyricinoleate (PGPR, food grade), bacterial cellulose (BC), and Nile red (BR, 98%) were acquired from Yuanye biology company. Food grade soybean oil was purchased from Fulinmen company (China Oil & Foodstuffs Corporation, Shanghai, China) and used without further purification. Beeswax (BW, melting point between 46–60 °C) was purchased from Aladdin (Shanghai, China). Other chemicals, including sodium alginate (viscosity 200 ± 20 mpa.s), 2,3-diacetyl (>99%) were purchased from Macklin company (Shanghai, China). All solutions were prepared with deionized (DI) water purified using a Milli-Q purification system.

### 2.2. Preparation of Pickering Double Emulsions

Sodium alginate at a concentration of 0.5 g was dissolved in citric acid-sodium citrate buffer solution with a pH of 5.8, to prepare sodium alginate solution (0.5 wt%) as the internal water phase (W_1_). The oil phase was prepared by dissolving PGPR (2.0 wt%) and different mass ratios of beeswax (0 wt%, 2.0 wt%, 4.0 wt %, 6.0 wt%, and 8.0 wt%) in soybean oil and then stirring at 65 °C 600 rpm for 20 min. The external water phase (W_2_) contained BC solution (1.0 wt%). Before preparation, the internal water phase (W_1_), oil phase (O), and external water phase (W_2_) was heated at 65 °C to maintain the melting state of beeswax during the preparation of double emulsions. W_1_ and O were mixed at a ratio of 3:7 and then homogenized (IKA-T18 digital, Staufen, Germany) at 13,000 rpm for 2 min to obtain the primary emulsion. Subsequently, the BC solution was added to the primary emulsion at a ratio of 5:5 and homogenized at 9000 rpm for 2 min to acquire Pickering double emulsions (PDEs) with different mass ratios of beeswax. To observe the microstructure of the double emulsions, the oil phase was dyed with Nile red prior to the homogenization. At least three replicates were performed for each type of formulation.

### 2.3. Microstructure Observation

The microstructure observation of PDEs was visualized using an inverted fluorescence microscopy (Carl Zeiss, Axio Observer A1, Oberkochen, Germany) to understand the distribution of the dispersed phase and continuous phase. Before the preparation of double emulsions, the oil phase was dyed with Nile red that was pre-dissolved in ethanol. Samples were diluted and carefully deposited onto the concave glass slide to minimize possible destruction, then covered with a cover slide and imaged under the magnification of 100× or 200×. For samples that were difficult to dilute (6.0% and 8.0% BW), they were directly smeared onto a concave glass slide and covered with a cover slide, then observed at 100× to obtain the microstructure.

### 2.4. Rheological and Tribological Measurements

The effects of beeswax addition on the rheological and tribological properties of PDEs were measured by rheometer (HAAKE MARS, Thermo Fisher, Waltham, MA, USA). After preparation of double emulsions, they were left at room temperature for 12 h until beeswax cooled n, to form gelling Pickering double emulsions. Firstly, the linear viscoelastic region (LVR) of double emulsions was determined in strain-oscillating scanning mode, and the elastic modulus (G′), viscous modulus (G″), strain (γ) and stress (τ) and the crossover point were recorded [20]. For steady-state shear experiments, the shear rate was increased from 0.1 to 100 s^−1^ to collect the viscosity and analyzed the viscosity at a shear rate of 50 s^−1^ of double emulsions. The frequency sweep was performed in the range of 0.1–10 Hz, and the elastic and viscous moduli of the samples were recorded as a function of frequency. All the tests were performed at 20 °C three replicates were performed for each type of formulation.

Rheometer coupling with friction module was applied to determine the oral tribology of double emulsions with different mass ratios of beeswax [21,22]. Commercially polydimethylsiloxane (PDMS) ball and pins were used for the measurement. After installing the friction module and manually adjusting the zero point, about 0.6 mL samples were put into the measuring tank, exerted a constant normal load force of 2 N and the measurement temperature was set at 37 °C. After equilibrium, sample temperature was set for 5 min, the entrainment speed increased from 0.1 to 300 mm/s within 10 min and recorded the friction coefficient of double emulsions. All the measurements were repeated 3 times.

### 2.5. Determination of Stability during Storage and Processing

The turbiscan apparatus (Formulaction, Toulouse, France) was used to monitor the stability of Pickering double emulsions with different mass ratios of beeswax during storage and after different processing conditions according to the method described by Matos, et al. Matos, Gutiérrez, Martínez-Rey, Iglesias and Pazos [23]. After preparation of double emulsions, they were poured into a flat-bottomed glass cylindrical sample cell in the flowing state with the height of samples being 44 mm. The storage stability of double emulsions was determined by measuring backscattering (BS) and transmission (TS) profiles as a function of time and cell height at scheduled time points (0 h, 1 d, 3 d, 5 d, 7 d, 14 d) at 30 °C. Similarly, the stability of samples after heating in water bath (30 min, 65 °C), 3 cycles freeze–thaw (froze at −20 °C, thawed at 25 °C) and the osmotic stress (0.5 M NaCl in W_2_) was determined by measuring the BS and TS profiles, too. We carried out a single scan as a reference curve to compare the changes in stability among samples before and after treatment. During the sample test, the sampling interval was 3 min, and the total sampling time was 1 h. To investigate the dynamic instability of samples, the last BS curve was analyzed. In addition, the turbiscan stability index (TSI), the change of backscattering (∆BS%) at particular sampling position and the peak thickness of the samples was obtained by subtracting the initial curve (t = 0) from the subsequent ones.

### 2.6. Flavor Release Properties of Pickering Double Emulsions

During the preparation of double emulsions with different mass ratios of beeswax, 2,3-diacetyl (log p = −1.43, 20 µL) was added to the internal water phase (W_1_) to form flavored emulsions. After the high-speed homogenization, flavored emulsions (2.0 g) were transferred to 20 mL glass bottles and closed with the silicone/PTFE seals screw caps. Double emulsions were held at 20 °C and 37 °C for 6 h to ensure the balance of volatile release in emulsions and air. Subsequently, the electronic nose (PEN3, Schwerin, Germany) was used to analyze the air-emulsion partition coefficient at equilibrium for double emulsions. In order to further study the retention effect of double emulsions with different mass ratios of beeswax during a 7-days storage, the emulsions were stored at room temperature with the storage bottle opened. The retention of flavor in the double emulsions was measured at specific time points (1 d, 3 d, 5 d, and 7 d) by the electronic nose. Before 6 h of the specified time point, the flavored emulsions were covered with the silicone/PTFE seals screw caps to balance the flavor. Before the measurement, the machine was adjusted three times with air to ensure accuracy and stability. During the measurement, the sampling interval was 1.0 s, and the measurement time was 100 s. All the experiments were repeated three times.

### 2.7. Statistical Analysis

All the experiments were carried out in triplicates. Origin Pro 9.1 software was used to perform the statistical analysis. One-way analysis of variance (ANOVA) procedure followed by Duncan’s multiple-rage test was used to determine the significance analysis among mean values at *p* < 0.05 with SPSS 26.0. The results were reported as the mean standard ± deviation (SD).

## 3. Results and Discussion

### 3.1. Formation and Characterization of Double Emulsions

The inverted fluorescence microscope (Carl Zeiss, Axio Observer A1, Oberkochen, Germany) was used to study the microstructure of Pickering double emulsions with different mass ratios of beeswax. Figure 1A showed that with the increase in BW, the fluidity of double emulsions gradually decreased. When the content of BW reached to 6.0% and 8.0%, double emulsions formed self-stand gel. In Figure 1B, the oil droplets (red regions) contained a significant amount of internal water phase (W_1_), as evident by the darker unstained spherical regions of varying sizes within the oil droplets of double emulsions (with 0–4.0% BW). The “droplets in droplets” structure indicated that double emulsions still maintained the typical microstructure when BW was added at concentrations of 0–4.0%. In addition, the average droplet size (D_4,3_) of the double emulsions significantly increased from 19.20 µm to 134.05 µm when the BW increased from 0% to 4.0%. This suggested that the addition of BW resulted in a higher viscosity of the dispersed phase, making it difficult to obtain droplets with smaller average size during the second homogenization [2].

However, when the beeswax concentration reached 6.0%, there was a change in the shape of the droplets, but typical structure of double emulsions was still maintained. Additionally, as the mass ratio of BW reached 8.0%, most of the oil droplets coalesced and formed “banded” structures, and the typical “bubble in bubble” structure of the double emulsions was lost. Probably, higher content of beeswax led to higher viscosity of primary emulsion, the inhomogeneous secondary emulsification caused the aggregation of oil droplets [14,24]. Besides, the large size and poor fluidity of droplets might also lead to the coalescence of adjacent droplets during microscopic observation, and thus the formation of irregular “banded” structures. After 14 days’ storage at room temperature, double emulsions still maintained the typical “droplets in droplets” structure without obvious structural degradation, but the growth rate of droplet size in double emulsion with 0% BW was significantly faster than that with oleogelation of the oil phase (data not showed). Therefore, the mass ratios of BW exerted great impact on the formation and microstructure of double emulsions.

### 3.2. Effects of BW Addition on the Rheological and Tribological Properties of Double Emulsions

Rheological properties reflected the fracture stress and strain determined by the hardness and brittleness, while tribological properties characterized by the friction coefficient linked with the creaminess or smoothness of foods [25,26]. Rheological analysis is an ideal method to study the relationship between structure and function of double emulsions. The results of oscillatory strain sweeps (at a fixed frequency of 1 Hz) and frequency sweeps (at a fixed strain of 0.5%) for double emulsions were presented in Figure 2A,B**,** respectively. As can be seen inFigure 2A, the storage modulus (G′) was always significantly smaller than the loss modulus (G″) of double emulsions with 0% and 2.0% BW, suggesting that they exhibited primarily fluid behavior. As the beeswax reached 4.0%, G′ was larger than G″ in their individual linear viscoelastic region (LVR), illustrating that they displayed mostly solid-like behavior. For double emulsions with the mass ratios of BW of 4.0%, 6.0%, and 8.0%, the stress (τ) and viscoelastic modulus (G′ and G″) at the end of LVR increased with the increase in beeswax (Table 1). Similarly, the strain (γ) and stress (τ) at the crossover point also increased along with the increase in the mass ratios of beeswax. These results indicated that the resistance of double emulsions against deformation were improved with the increase of beeswax.

Moreover, as seen from Figure 2B, the G″ > G′ appeared in double emulsions with 0% and 2.0% BW, while the G′ > G″ with 4.0–8.0% BW, which further confirmed the results in Figure 2A. Both G′ and G″ for all samples displayed a relatively strong frequency dependence, suggesting that the rheological response of double emulsions was largely influenced by the applied deformation rate at frequency from 0.1 to 10 Hz. The apparent viscosity of all the double emulsions decreased with the increasing shear rate (Figure 2C), which was the characteristic of soft materials with the shear thinning behavior. Moreover, the addition of BW lifted the apparent viscosity at any shear rate, which was due to the stronger resistant to deformation after gelation oil phase. The apparent viscosity at a shear rate of 50 s^−1^ was found to have a significant effect on the perception of smoothness during oral processing [25,27]. We thus analyze the viscosity at a shear rate of 50 s^−1^, wherein the viscosity of double emulsions differed greatly with different mass ratios of beeswax, especially the double emulsion with 0% and 2.0% BW. This indicated that the addition of BW in oil phase and the strength of gelation could indeed affect the oral sensation of double emulsions.

To further explore the impact of oil phase gelation on the mouthfeel of double emulsions, the friction coefficient was measured. The oral tribology determined the mouthfeel when food and/or food-saliva mixtures interact with the oral surface during the later stages of oral processing [28,29]. As shown in Figure 2D, the friction coefficient of double emulsions decreased along with the increase in BW content that indicates a smoother mouthfeel. It was reported that when the droplets entered the contact zones, the droplets with larger particle size produced higher deformability, which formed a viscous film or patches of oil film in the tribo-surfaces [21]. This might explain the reduction in friction coefficient at higher content of BW, wherein larger droplet size was obtained. Besides, the human tongue speeds had been estimated to be between 10 and 200 mm/s and the entrainment speed at 10 mm s^−1^ was a typical speed during oral processing [27,30,31]. Taking into account the relationship between the mouthfeel and tribological measurements was of great significance, particularly at low speed (e.g., 10 mm s^−1^). In Figure 2D, the concentration of BW affected the friction coefficient at 10 mm s^−1^ (µ_10_) significantly. As the BW increased, µ_10_ decreased, suggesting that the addition of BW improved the lubrication property and smooth attributes of double emulsions. Therefore, the oral sensation of double emulsions could be adjusted by controlling the beeswax content.

### 3.3. Effects of BW Addition on the Storage Stability of Double Emulsions

To realize a direct, quick, objective, and quantitative evaluation of dispersions, Turbiscan, an efficient method was used for evaluating the stability during long-term storage and under different processing conditions of double emulsions [32]. Figure 3A showed the backscattering (BS) profiles of the samples after 14 days of storage, which was a function of the sample height, characterized the homogeneity to obtain a stable or unstable fingerprint of double emulsions. As depicted in Figure 3A, an increase in ∆BS at the middle or top height and a decrease in ∆BS at the bottom height suggested that oil droplets migrated from the uniform stage to the top layer, revealing creaming process during storage. In double emulsions with 0% and 2.0% BW, we also discovered the ∆BS decreased at the top, this indicated that oiling-off phenomenon had occurred. Apart from the different density between water and oil, the flocculation, aggregation, and coalescence of droplets accelerated the creaming, even oiling-off of the double emulsions.

Then, we analyzed the mean change of ∆BS at 0–20 mm and the clarification layer (peak thickness) of double emulsions with different concentrations of BW in Figure 3B. The mean change of ∆BS represented the change of droplets concentration in a specific region of the sample, the larger the value, the change in droplets concentration was more significant. It was important to point out that the mean change of ∆BS and the peak thickness were reduced along with the increase in BW content. This may be due to the increase of viscosity and postponed the migration of droplets after gelation with BW [33]. In addition, the stability of double emulsions was evaluated using the Turbiscan stability index (TSI), and the results were shown in Figure 3C,D. The TSI was also utilized to assess the destabilization kinetics verse the storage time [34,35]. When TSI < 3.0, it means that no visible destabilization occurred, and TSI > 3.0, the creaming or sediment was noticed. With the increasing mass ratios of BW, the TSI values decreased as expected, especially the content of BW at 6.0% and 8.0%, the TSI < 3.0, which implied that a stronger gelation strength had a positive effect on the long-term stability of double emulsions at room temperature. The TSI value at the top could effectively reflect the instability of the samples at the top, especially the degree of creaming or oiling-off. As shown in Figure 3D, the TSI value of the double emulsion with 0% BW was apparently higher than that of the double emulsions with oil phase gelation. This further demonstrated the improvement effect of oil phase gelation on the long-term storage stability of double emulsions.

### 3.4. Effects of BW Addition on the Processing Stability of Double Emulsions

Heating, freeze–thaw, and osmotic pressure treatments are among the commonly encountered processing conditions during food processing, and it is therefore important to know the stability of double emulsions after these treatments for guiding the production and application of double emulsions. The stability of double emulsions after heating in water bath for 30 min at 65 °C was displayed in Figure 4. An increase in ∆BS at the top and a decrease at the bottom of the sample indicated that an obvious creaming process occurred for the oil droplets after the heating treatment. All the double emulsions showed an obvious clarifying layer at the bottom in Figure 4A, while for double emulsion without the addition of BW apparent oiling-off phenomenon even appeared at the top. Subsequently, we compared the peak thickness and found that double emulsions with 0% BW had the largest peak thickness, up to 33.72 mm, while there was no significant difference in peak thickness among samples gelled with different mass ratios of beeswax (Figure 4B). The change in the thickness of the clarifying layer or the oiling-off layer indicated that the destabilization took place by droplets flocculation, coalescence or Ostwald ripening phenomena [23,36]. The average change in ∆BS at the top (Figure 4C) also confirmed the occurrence of droplets migration from bottom to top. The highest mean change of ∆BS in sample with 0% BW was not only related to droplets migration, but also due to the lower backscattering light of oil phase compared with the emulsion phase caused by oiling-off. However, in Figure 4D, the double emulsion with 8.0% beeswax had the highest TSI indicating it was the most unstable after heating, probably due to the melting of beeswax crystals at the interface or in the continuous phase. Therefore, gelation oil phase could not effectively improve the heating stability of double emulsions, the melting of beeswax crystals accelerated the instability of double emulsions during heating.

Understanding the influence of freeze–thaw treatment on the stability of double emulsions is of great significance for applying double emulsion in foods requiring cold chain transportation [37]. It could be seen from the backscattering curves (Figure 5A) that all double emulsions experienced the creaming process, but with no oiling-off phenomenon. The clarifying layer also appeared in double emulsions loaded with 0% to 4.0% BW. We further compared the peak thickness of double emulsions with different mass ratios of BW (Figure 5B) and found that with the increase in beeswax content, the peak thickness of the samples decreased gradually. In addition, the peak thickness of double emulsions after freeze–thaw was lower than that of heating under the same mass ratio of beeswax. Surprisingly, the peak thickness of double emulsion with 8.0% BW dropped to zero, suggesting that the double emulsion could still maintain the superior stability even after 3 cycles of freeze–thaw treatments. The mean change of ∆BS at 0–20 mm (Figure 5C) also proved that the change in droplets concentration at the bottom (which indicates the migration of droplets) decreased along with the increase of beeswax in double emulsions. In Figure 5D, the TSI of the double emulsion with 0% BW reached nearly 20.0 (a sample with TSI > 10.0 was considered as severe unstable). After gelation of oil phase with beeswax at 2.0%, the TSI rapidly decreased by about 50%, and with the increase of beeswax, the TSI progressively decreased. When the mass ratios of beeswax equaled 6.0% or 8.0%, a TSI value less than 3.0 was obtained, meaning that the double emulsions had no visible instability after 3 cycles of freezing-thaw. The instability caused by freeze–thaw in emulsions was therefore mainly attributed to the formation of ice crystals during freezing which then occupied the position of oil droplets or penetrated through the interface film and eventually led to a series of unstable phenomena. The beeswax crystals at interfacial or oil phase could effectively resist the instability caused by ice crystal formation. Besides, gelation of oil phase that formed thicker interfacial membranes and increased the viscosity were more effective in preventing partial coalescence [19], and consequently improved the freeze–thaw stability of double emulsions.

Osmotic pressure existing between the internal and external aqueous phase results in swelling (and eventual rupture) or coalescence of droplets, leading to instability of double emulsions. Additionally, during the processing or storage, the dilution or shearing may also arise in the osmotic stress of double emulsions [1,11]. Therefore, we added 0.5 M NaCl in the external water phase and studied the effect of different mass ratios of beeswax on the resistance of double emulsions to osmotic pressure. It can be seen that creaming occurred in double emulsions with 0–4.0% BW, but double emulsions with 6.0% and 8.0% BW still remained smooth curves, indicating that the droplets migration and the change in droplet size in emulsions with 6.0% and 8.0% BW were minor under osmotic pressure stress (Figure 6A). Similarly, the mean change of ∆BS at the bottom and the peak thickness of the whole samples decreased with the increase of the beeswax. Stability was also evaluated through TSI value evolution over a period of 1 h (Figure 6C). The TSI value of double emulsions with 0% BW gradually increased to 15.91 during the whole test period. When the beeswax increased to 2.0%, the TSI value decreased to 10.39. Additionally, the TSI was less than 3.0 when the mass ratio of beeswax fell in 4.0–8.0%, no visible instability or no significant variation occurred. Besides, the change of backscattering light in the middle height of the sample represents the evolution of droplet size, and the larger change symbolized the more remarkable increase in the droplet size [32]. As shown in Figure 6D, the ∆BS at 30 mm of double emulsions with 0% and 2.0% BW varied greatly during the testing process, especially the 0% BW, indicating that the droplets had obvious flocculation or coalescence under the treatment of osmotic pressure. Whereas the ∆BS at 30 mm in double emulsions with 4.0–8.0% BW changed little. Therefore, the results demonstrated that the resistance of double emulsions to osmotic pressure could be increased by forming a fat crystal network within the oil phase to immobilize the oil droplets [38]. Moreover, gelation of the oil phase inhibited the water movement and also affected the film fusion and solute exchange processes of droplets to improve the stability of double emulsions. Therefore, the presence of BW as well as the increase in mass ratios did not significantly improved the heating stability, but the freeze–thaw and osmotic pressure stability of the double emulsions was ameliorated.

### 3.5. Effects of BW Addition on the Flavor Release Profile of Double Emulsions

Flavor is one of the most important criterion to determine the consumer acceptance of food products, especially for double emulsions with fat reduction function. The effects of different mass ratios of BW on the flavor release of double emulsions were shown in Figure 7. It was obvious that the air-emulsion partition coefficients at equilibrium of double emulsions differed significantly at 20 and 37 °C, and the partition coefficient decreased with the increase in BW content. Double emulsion with 0% BW, there was no significant difference in the partition coefficients at different temperature, which may be because the oral processing temperature had no great influence on the microstructure, including the properties of phase (water and oil) and interface and the mechanical properties of double emulsion. When the oil phase was gelled with 2.0% and 4.0% BW, the air-emulsion partition coefficient was larger prominently at 37 °C than 20 °C (*p* < 0.05), due to the partial melting of beeswax crystals at the interface or oil phase that facilitated the mobility and diffusion of flavor [16,39]. However, when the mass ratio of beeswax increased to 6.0% and 8.0%, the air-emulsion partition coefficient had no significant difference at 20 °C and 37 °C (*p* > 0.05), suggesting that higher content of beeswax formed more compact fat crystal networks, which were more resistant to partial melting caused by oral processing temperature.

In addition, the dynamic flavor release of double emulsions during a period of one week was monitored by an electronic nose (Figure 7B). The response value of double emulsions decreased with the extension of exposure time, indicating that the flavor was constantly released into the air during the opening storage. For the double emulsion without gelation, its response value dropped sharply on the first day of opening storage, with a decrease of 51.04%, showing a burst release behavior. Surprisingly, the burst release of flavor in double emulsions was inhibited obviously after gelation of oil phase with BW. Besides, the decreasing amplitude of the response values of double emulsions with different mass ratios of BW differed significantly after 7 days’ of opening storage. The decreasing amplitude of double emulsion with 0% beeswax was 73.08 ± 0.79%, much larger than that of samples with the gelled oil phase (*p* < 0.05). While the double emulsion with 8.0% BW, it manifested a slow and sustained release, with the smallest decreasing amplitude of 28.21 ± 0.04%. This may be due to the steric barrier formed by the beeswax crystals at the oil droplets interface and the higher gel strength of double emulsions with higher beeswax content [40,41]. Therefore, a modification of emulsion microstructure and mechanical properties by regulating the content of BW may hinder the release of flavor and achieve sustained or delayed release of volatiles during consumption and storage.

## 4. Conclusions

We prepared Pickering double emulsions gelled with beeswax and the effects of different mass ratios on the formation, stability, oral sensation, and flavor release properties were evaluated. The addition of beeswax influenced the droplet size and microstructure of double emulsions. It was possible to adjust the mechanical and tribological properties of double emulsions by altering the BW content. With the increase in beeswax, double emulsions manifested stronger resistance over deformation and contributed to a smoother mouthfeel. Additionally, oleogelation of the oil phase with beeswax improved the colloidal stability of double emulsions according to the BS curve and TSI during a long-term storage, freeze–thaw, and osmotic pressure treatments. The stability of double emulsions against environmental stresses was positively dependent on the mass ratios of beeswax. However, the heating stability of double emulsions was not ameliorated effectively. We also found that the air-emulsion partition coefficients at equilibrium of double emulsions was regulated effectively by the mass ratios of beeswax. Besides, the gel network strengthened by beeswax inhibited flavor release during the opening storage. Therefore, gelation with beeswax could be useful in the formulation of double emulsions with designable stability, texture, and flavor release profile for a better oral perception. The findings of the research would be useful in designing emulsion-based foods with improved physicochemical, mechanical, and functional attributes. Knowledge obtained in this study will facilitate the development of double emulsions with designable flavor profile and offers guidance for designing low-fat emulsions with improved oral perception.

## Figures and Tables

**Figure 1 foods-11-01039-f001:**
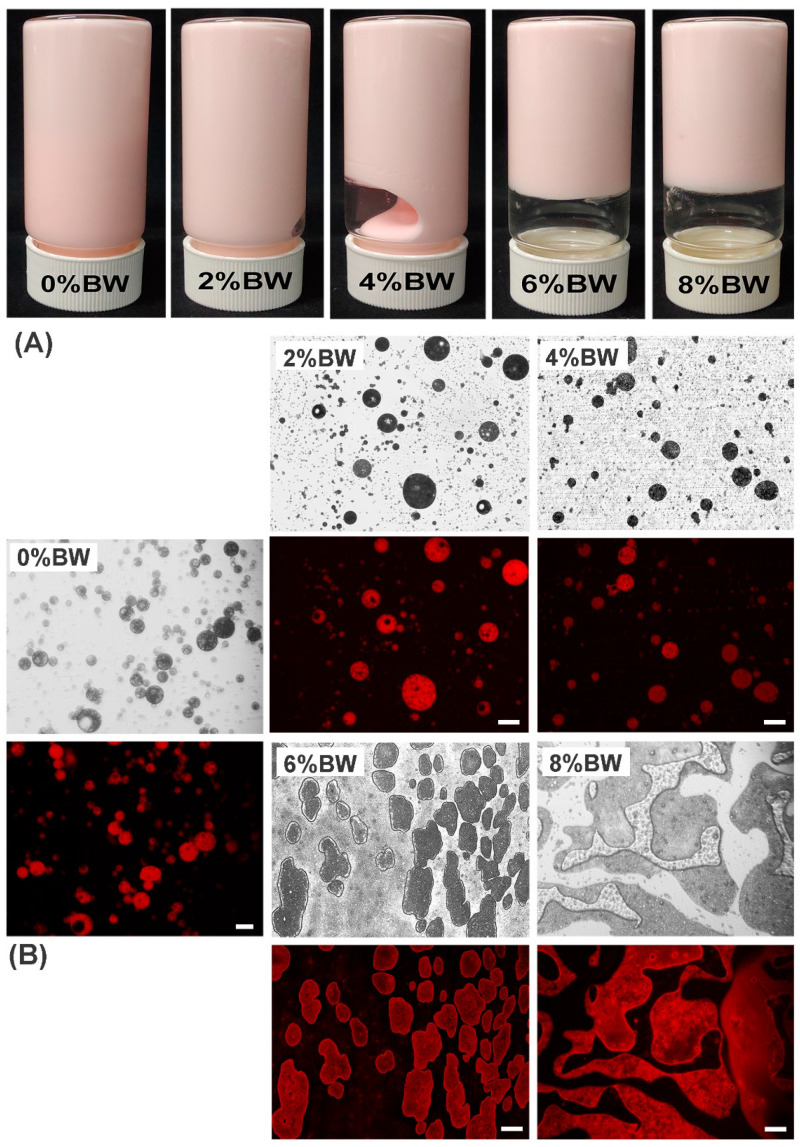
The digital (**A**) and optical microstructure (**B**) images of Pickering double emulsions with different mass ratios of beeswax in the oil phase. The scale bar of the double emulsion with 0 wt% BW was 20 µm and the others were 100 µm.

**Figure 2 foods-11-01039-f002:**
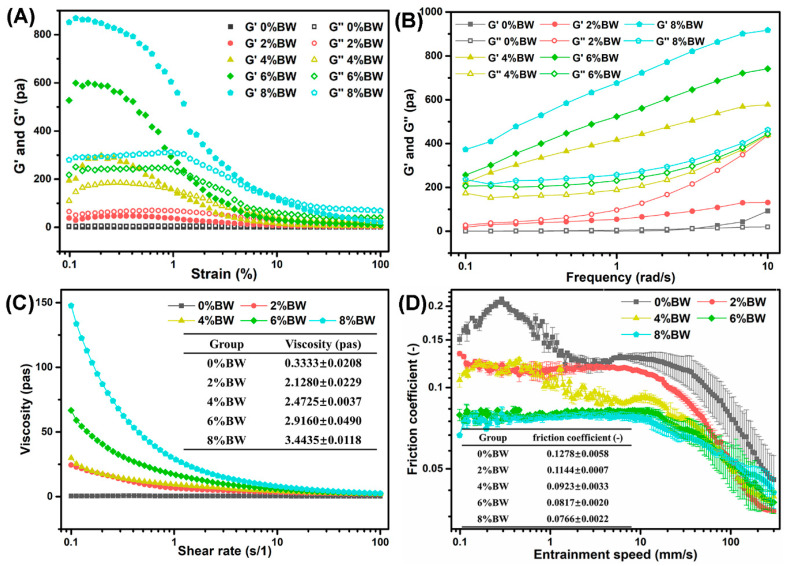
The influence of beeswax addition on the rheological and tribological properties of double emulsions: (**A**) the amplitude strain sweeps; (**B**) the frequency sweeps; (**C**) the steady shear flow measurements; and (**D**) the friction coefficients.

**Figure 3 foods-11-01039-f003:**
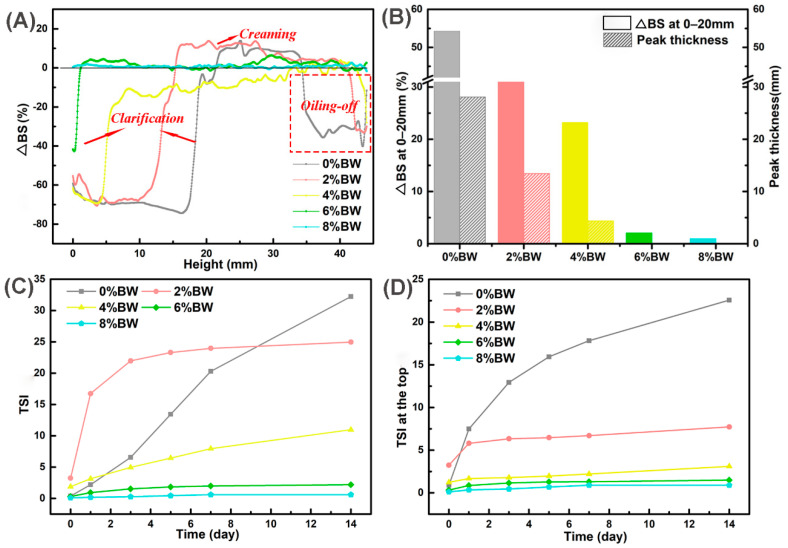
The stability of double emulsions with different mass ratios of beeswax during long-term storage at room temperature: (**A**) the last backscattering profile after 14 days of storage; (**B**) the mean change of backscattering (∆BS%) at 0–20 mm and the peak thickness of the sample after 14 d, the gray, red, yellow, green and blue color represented double emulsions with 0%, 2%, 4%, 6% and 8% BW; (**C**) the TSI values of the sample at different time intervals; and (**D**) the TSI values at the top (30–44 mm) of the sample at different time intervals.

**Figure 4 foods-11-01039-f004:**
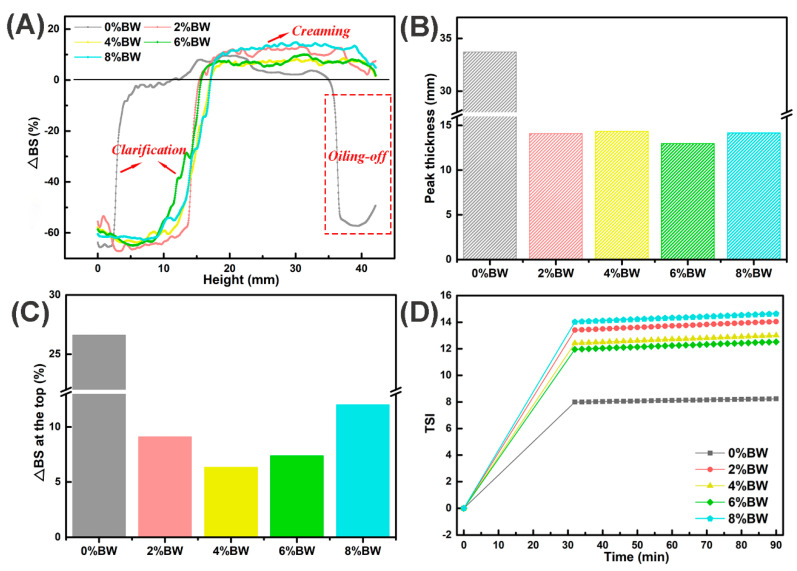
The processing stability of double emulsions with different mass ratios of beeswax after heating at 65 °C for 30 min: (**A**) the last backscattering profile after heating treatment; (**B**) the peak thickness of the sample after heating treatment, the gray, red, yellow, green and blue color represented double emulsions with 0%, 2%, 4%, 6% and 8% BW; (**C**) the mean change of ∆BS at the top (30–44 mm) of the sample after heating treatment, the gray, red, yellow, green and blue color represented double emulsions with 0%, 2%, 4%, 6% and 8% BW; and (**D**) the TSI curves after heating treatment.

**Figure 5 foods-11-01039-f005:**
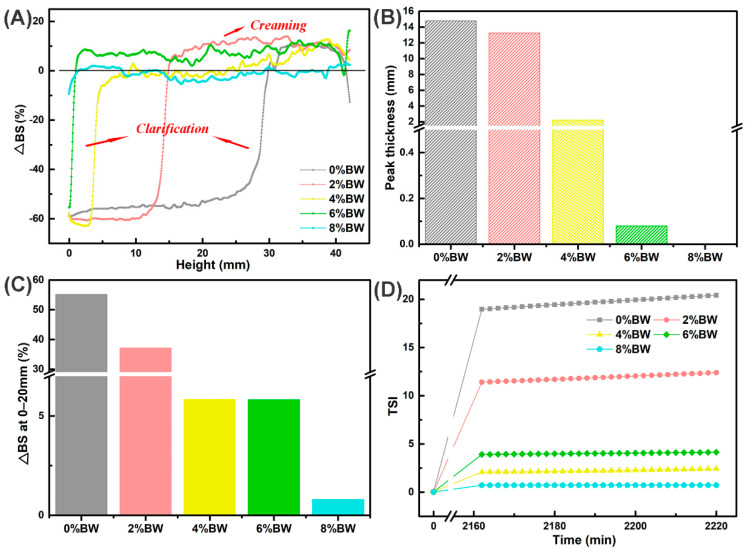
The processing stability of double emulsions with different mass ratios of beeswax after 3 cycles of freeze–thawed: (**A**) the last backscattering profile after 3 cycles of freeze–thaw treatment; (**B**) the peak thickness of the sample after 3 cycles of freeze–thaw treatment, the gray, red, yellow, green and blue color represented double emulsions with 0%, 2%, 4%, 6% and 8% BW; (**C**) the mean change of ∆BS at the 0–20 mm of the sample after 3 cycles of freeze–thaw treatment, the gray, red, yellow, green and blue color represented double emulsions with 0%, 2%, 4%, 6% and 8% BW; and (**D**) the TSI curves after 3 cycles of freeze–thaw treatment.

**Figure 6 foods-11-01039-f006:**
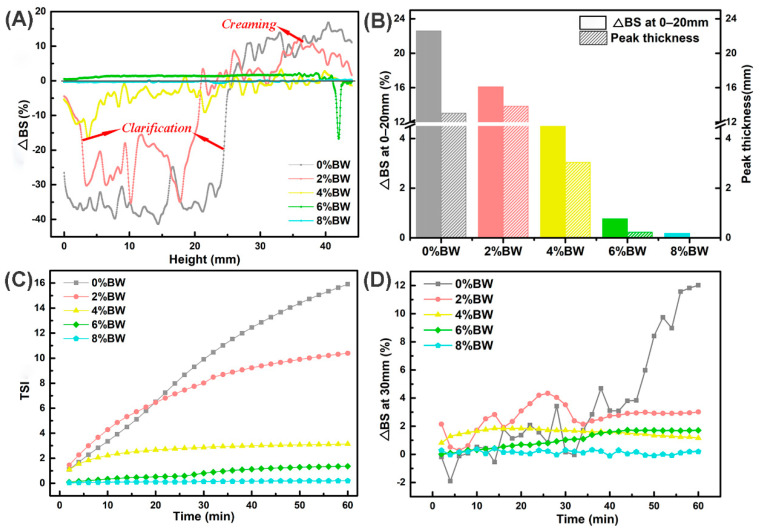
The processing stability of double emulsions with mass ratios of beeswax after osmotic stress (0.5 M NaCl at W_2_): (**A**) the last backscattering profile after osmotic stress treatment; (**B**) the mean change of ∆BS at 0–20 mm and the peak thickness of the sample after osmotic stress treatment, the gray, red, yellow, green and blue color represented double emulsions with 0%, 2%, 4%, 6% and 8% BW; (**C**) the TSI curves after osmotic stress treatment; and (**D**) the change of ∆BS at 30 mm after osmotic stress treatment.

**Figure 7 foods-11-01039-f007:**
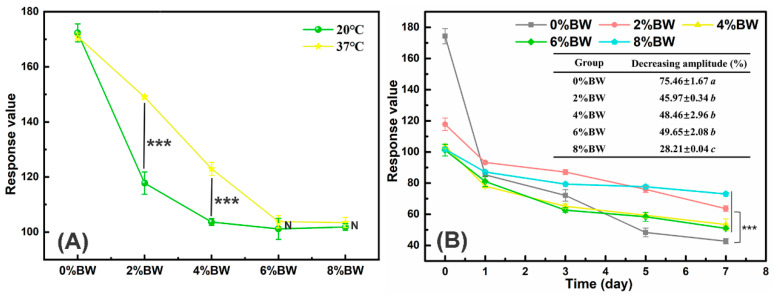
The flavor release of double emulsions with different mass ratios of beeswax: (**A**) the air-emulsion partition coefficients at equilibrium at different temperature; (**B**) the flavor release profile during open-cap storage. “***” indicated the significant differences among samples (*p* < 0.001) and “N” represented no significant differences between samples, the lower case letters (a–c) indicated significant differences (*p* < 0.05) in the decreasing amplitude (%) between double emulsions with different BW concentrations.

**Table 1 foods-11-01039-t001:** The effect of concentration of beeswax on the end of linear viscoelastic region (LVR) and crossover point of double emulsions.

Group	End of LVR			Crossover Point		
	Τ (pa)	G′ (pa)	G″ (pa)	Γ (%)	Τ (pa)	G′ = G″ (pa)
4.0% BW	2.12 ± 0.03	159.80 ± 9.72	158.70 ± 1.71	0.10 ± 0.00	0.18 ± 0.01	125.50 ± 3.43
6.0% BW	2.34 ± 0.08	519.90 ± 16.94	241.40 ± 6.74	1.38 ± 0.06	4.30 ± 0.15	221.40 ± 1.80
8.0% BW	4.12 ± 0.19	758.40 ± 7.27	304.90 ± 1.10	7.98 ± 0.36	15.22 ± 0.05	135.50 ± 5.59

## Data Availability

The data that supported the findings of this study were available within the article.

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
