# Peer review of "Regulation Effects of Beeswax in the Intermediate Oil Phase on the Stability, Oral Sensation and Flavor Release Properties of Pickering Double Emulsions"

_foods, 2022, doi:10.3390/foods11071039_

Round 1

Reviewer 1 Report

The manuscript aimed to produce Double emulsions with oil intermediate phase gelled with beeswax. The effect of beeswax mass ratios (0 - 8%) on the stability, rheology, oral sensation, and flavor release profile of the emulsions were investigated. The topic addressed in this paper is relevant and it contains interesting results and discussions; however, some questions need to be solved.

1. When the beeswax concentration reached 6%, the double emulsion structure was maintained according to microscopy images, but in this concentration, there was a change in the shape of the double emulsion droplets. However, when the beeswax concentration reached 8%, the structure of the double emulsion was almost wholly lost. It is possible to see on the microscopy images that most of the system is a simple O/W emulsion. This change in the structure of emulsions also can affect the results of rheology, tribology, and especially stability. I think that it should be raised during the discussion of the results.

2. How did you calculate the TSI values at the top of the samples (Figure 3D) at different time intervals? In this case, did you analyze only the destabilization at the top of the measuring cell?

3. How could you prove that there was oleogelation of the oil phase?

4. Line 423 – “droplets size” – English mistake. Check all text.

Reviewer 2 Report

A brief summary

The article „Regulation effects of beeswax in the intermediate oil phase on 2 the stability, oral sensation and flavor release properties of 3 Pickering double emulsions“ is focused on the assessment of stability and characterization of w/o/w double emulsions stabilized by beeswax. This method of stabilization is original and provides the improvement of specific properties of double emulsions (stability in general, controlled flavor release etc.).

General concept comments

The idea of double emulsion stabilitazion in this article is innovative but I think there could be some unpleasant sensory properties for potential consumers of these products. However, this article is a basic research. To highlight the good stability of measured samples, authors should provide more results obtained during the selected storage period – at least at the end of storage.

Specific comments:

Abstract

-line 10: for better understanding - specify the kind of prepared double emulsions: W/O/W

Introduction

-line 72: Are there any recommendations for maximum content of beeswax in food?

Materials and Methods

-line 96: How was the speed of mixing?

-line 99 – 102: Which kind of stirrer was used? Add the instrument specification …

-line 108: Was the dye added to all samples? It could be useful to mention this information in „2.2 Preparation of Pickering double emulsions“.

-line 116: Why did you not repeat these measurements during storage – e.g. at the end of storage period?

-line 136: How was the temperature of storage?

-line 144 + line 153: similar to line 108

Results and discussion

-line 184: Did you measure the droplet sizes by microscope? Did you also evaluate these microstructures of double emulsions during storage – e.g. at the end of storage period?

-line 273 (Figure 3D): It might be useful to specify „the top“ in mm

Figure 4, 5, 6, 7A: Were these analyses performed after the preparation of samples? Why did you not repeat these measurements during storage?

Reviewer 3 Report

Whom corresponds the method of friction module to determine the oral tribology?;

Figure 2A and Table 1: The strain must be in units of Pa;

Figures 3, 4,5 and 6: The units and figures must be better discussed;

Figure 7: Which is the unit of response value?;

In figures and tables must be added standard deviation.

Reviewer 4 Report

The authors studied the potential use of beeswax as a gelater to prepare double emulsion gel with improved stability. They investigated the effects of different mass ratios of beeswax on the microstructure of double emulsion gels. They also the stability and the release behaviour of the system. The work is well-written and presented and of interest to the food industry. 

Author Response

Thanks to reviewers for the positive comments.